# Antimicrobial Activity of Peptides Produced by *Lactococcus lactis* subsp. *lactis* on Swine Pathogens

**DOI:** 10.3390/ani13152442

**Published:** 2023-07-28

**Authors:** Fernando M. M. Sanca, Iago R. Blanco, Meriellen Dias, Andrea M. Moreno, Simone M. M. K. Martins, Marco A. Stephano, Maria A. Mendes, Carlos M. N. Mendonça, Wellison A. Pereira, Pamela O. S. Azevedo, Martin Gierus, Ricardo P. S. Oliveira

**Affiliations:** 1Department of Biochemical and Pharmaceutical Technology, School of Pharmaceutical Sciences, University of São Paulo, São Paulo 05508-900, Brazil; fernando.mamani.ms@gmail.com (F.M.M.S.); iagoblanco@usp.br (I.R.B.); meriellend@gmail.com (M.D.); well.ap@usp.br (W.A.P.); pam.o.souza@usp.br (P.O.S.A.); 2Dempster Mass Spectrometry Laboratory, Chemical Engineering Department, Polytechnic School, University of São Paulo, São Paulo 05508-000, Brazil; mariaanita.mendes@gmail.com; 3Department of Preventive Veterinary Medicine and Animal Health, School of Veterinary Medicine and Animal Sciences, University of São Paulo, São Paulo 05508-000, Brazil; morenoam@usp.br; 4Department of Animal Sciences, Faculty of Animal Sciences and Food Engineering, University of São Paulo, São Paulo 05508-000, Brazil; smmkm@usp.br; 5Immunobiological and Biopharmaceutical Laboratory, School of Pharmaceutical Sciences, University of São Paulo, São Paulo 05508-000, Brazil; stephano@usp.br; 6Institute of Animal Nutrition, Livestock Products, and Nutrition Physiology (TTE), Department of Agrobiotechnology, IFA-Tulln, University of Natural Resources and Life Sciences, 1190 Vienna, Austria; martin.gierus@boku.ac.at

**Keywords:** bacteriocin, lactic acid bacteria, pig, *Lactococcus lactis*, peptides

## Abstract

**Simple Summary:**

Antimicrobial peptides naturally produced by probiotic bacteria are used as alternatives to the long-term use of antibiotics against pathogenic bacteria. In the present study, we tested the effect of secreted compounds produced by the probiotic bacterium *Lactococcus lactis* subsp. *lactis* strain L2 on inhibiting the growth of pig pathogens in vitro. It was shown to be effective by bacteriostatic and bactericidal mechanisms in a strain-dependent manner, particularly against Gram-positive bacteria. The present analyses indicate that the molecules secreted in the cell-free supernatant responsible for this effect are of very low molecular weight, which may be responsible for this antimicrobial effect. Here we present potential beneficial effects of the use of probiotics in the control of pathogens in the pig industry.

**Abstract:**

Swine production is of great importance worldwide and has huge economic and commercial impact. Due to problems with bacterial infection, the use of antimicrobials has increased in the last decades, particularly in Latin America and Asia. This has led to concerns about antimicrobial resistance, which poses risks to human health and the environment. The use of probiotic organisms has been proposed as an alternative to this use, as these beneficial bacteria can produce antimicrobial peptides, such as bacteriocins, which allow the induction of inhibitory effects against pathogenic microorganisms. Among probiotics, some bacteria stand out with the inhibition of animal pathogens. The bacteriocin-like inhibitory substances (BLISs) of *Lactococcus lactis* subsp. *lactis* strain L2, present in its cell-free supernatant, were tested against pathogenic strains isolated from pig samples, such as *Escherichia coli*, *Salmonella enterica*, *Streptococcus suis*, *Streptococcus dysgalactiae*, *Staphylococcus hyicus*, and *Enterococcus faecalis*. Compounds secreted by *L. lactis* L2 have been shown to inhibit the growth of some pathogenic species, particularly Gram-positive bacteria, with *S. suis* being the most prominent. Antimicrobial peptides with a molecular size of 500–1160 Daltons were isolated from BLISs. The results highlight the potential of *L. lactis* BLISs and its peptides as natural antimicrobials for use in the food industry and to reduce the use of growth promoters in animal production.

## 1. Introduction

The profitability of intensive pig farms is directly related to the reproductive (litter/sow/year) and productive (weaner/sow/year) efficiency of the herd [1]. To improve animal production, it is important to consider the advances in management, genetic improvement, animal welfare, sanitary conditions, and nutrition [2]. The growing demand for animal protein, combined with concerns about economic and environmental sustainability, has led to the search for tools to make animal production more efficient [3].

One of several ways to do this would be to improve the health and nutrient utilization capacity of the pig. For this end, antibiotics have been widely used as a way to control the population of pathogenic bacteria in pig farms, such as *Salmonella* [4], *Clostridium* [5], *Streptococcus* [6], *Enterococcus* [7], and *Escherichia* [8]; thus, allowing the animals to demonstrate their genetic potential. However, their use as growth promoters has been questioned due to the development and spread of antimicrobial-resistant bacteria, which is a serious public health problem related to the transfer of resistance genes between animals and humans [9,10]. In addition, this use has also led to an imbalance in the gut microbiota of the piglet, which could affect later growth performance [11].

The use of probiotics in livestock has increased as an alternative to limiting the use of antimicrobials as growth promoters. Their effect has been linked to competitive exclusion, bacterial antagonism, and stimulation of the immune system by antibacterial and bacteriostatic substances such as organic acids, bacteriocins, diacetyl, and hydrogen peroxide [12]. Various antimicrobial peptides naturally produced by bacteria have been used to inhibit the growth of various pathogens, such as bacteria, parasites, viruses, and fungi [13,14,15]. These peptides can be naturally produced by bacteria of different genera, particularly lactic acid bacteria (LAB) such as *Bacillus*, *Pediococcus*, *Leuconostoc*, *Lactobacillus*, and *Lactococcus* [15]. Recently, new techniques have been applied to evaluate the bacterial ability to produce bacteriocins as alternative treatments to various diseases [15,16]. Bacteriocins can be classified into different classes based on their structure, molecular weight (MW), and genetic sequences, such as class I (lantibiotics), class II (heat-stable), and class III (heat-labile) [17], and when not fully characterized, the term bacteriocin-like inhibitory substance (BLIS) is recommended [18,19,20].

It has been reported that bacteriocinogenic *Lactococcus lactis* can exert an inhibitory effect against several pathogenic bacteria, including *Streptococcus agalactiae*, *Staphylococcus aureus*, *Listeria monocytogenes*, and *Salmonella enterica* serovar Choleraesuis, demonstrating a significant potential of these LAB in inhibiting pathogenic bacteria [21,22]. Therefore, it is known that benefits of probiotics, although host-specific, can be applied in different animal models [23,24]. Therefore, the aim of this study was to verify the antimicrobial capacity of *L. lactis* subsp. *lactis* strain L2 on pathogenic strains isolated from pigs. We hypothesized that the antimicrobial capacity of this strain is efficient against swine pathogens, highlighting its potential application in pig farming.

## 2. Materials and Methods

### 2.1. Bacterial Strains

*L. lactis* subsp. *lactis* strain L2 was provided by the Laboratory of Microbial Biomolecules of the University of São Paulo. The isolation and complete characterization of this strain was performed by Pereira et al. [21]. Pathogenic bacterial strains were isolated from pigs (*Sus scrofa domesticus*) reared in Brazilian swine herds, as part of the bacterial collection of the Swine Health Laboratory (FMVZ-USP). Identification of the isolates was confirmed by MALDI-TOF mass spectrometry. Information on these strains is given in Table 1.

### 2.2. Bacterial Growth Conditions

*L. lactis* subsp. *lactis* strain L2 was grown in 4 mL of De Man, Rogosa, and Sharpe (MRS) medium (BD Difco™, Strasbourg, France) at 37 °C for 24 h in a microaerophilic environment, where cultures were maintained in an anaerobic jar, using a candle to eliminate internal oxygen. After 24 h, 100 µL of the cell suspension was subcultured in 5 mL of MRS medium at 37 °C in a microaerophilic environment (anaerobic jar) for 18 h. The optical density at 600 nm (OD_600nm_) was measured at the beginning and end of the incubation. The culture suspension was centrifuged at 13,000× *g* for 10 min at 4 °C to obtain the cell-free supernatant [21], referred to as BLIS. The pH of *L. lactis* L2 BLIS was adjusted to 6.0–6.5 with NaOH (1M). Furthermore, the BLIS was kept at 80 °C for 10 min, and frozen at −20 °C until use. The pathogenic strains *E. coli* (Sta+, STb+, LT+, F4+, 987P+), *S. enterica* serovar Choleraesuis, *S. enterica* serovar Typhimurium, *S. suis*, *S. dysgalactiae*, *E. faecalis*, and *S. hyicus* were grown in Brain Heart Infusion (BHI) medium (KASVI™, Madrid, Spain) at 37 °C for 24 h in aerobiosis (BOD incubator; Bio-Oxigen Demand). *C. perfringens* was grown in thioglycolate medium (KASVI™, Madrid, Spain) at 37 °C for 24 h and stored in an anaerobic jar using a GasPak™ EZ Anaerobe Container System (BD™, Franklin Lakes, NJ, USA).

### 2.3. Antimicrobial Activity Assay

#### 2.3.1. Agar Diffusion

The concentration of pathogenic strains in culture was adjusted until reaching OD_600nm_ 0.2–0.3, corresponding to 3.0–4.0 × 10^8^ CFU/mL. Then, each adjusted concentration of pathogen was added to the agar medium (17.5 mL) at 1% (*w*/*v*) and poured onto Petri dishes (90 × 115 mm) until solidified. Wells of uniform diameter were marked on the solidified agar media, in triplicates for all pathogen strains. In each well, 80 µL of BLIS, sterilized through filters (0.2 µm) after pH correction, was allowed to diffuse on the plates for 2 h at room temperature, which were further incubated under optimal growth conditions (37 °C). The inhibition halos generated after 24 h were measured using a digital pachymeter [25,26] poured in mL (Formula (2)) [27].
*Ai* = (*Rt*)^2^ × π − (*Ro*)^2^ × π(1)
(2)$$\frac{AU}{ml}=\frac{Ai}{V}$$

In these formulas, Ai represents the area of inhibition around the well (cm^2^), Rt represents the total radius on the plate, while Ro represents the radius of the well. Quantification in arbitrary units per milliliter ($\frac{AU}{ml}$), shown in Equation (2), is calculated as the ratio of the area of inhibition to the volume (V) of BLIS added to each well. The antimicrobial activity is expressed in arbitrary units per milliliter (AU/mL) according to Formula (2).

#### 2.3.2. Growth Inhibition Test

An absorbance microplate reader (BioTek Synergy HTX, Winooski, VT, USA) was used to assess the turbidity of pathogen strain cultures exposed to *L. lactis* L2 BLIS, and the absorbance was used to calculate bacterial inhibition growth using Gen5 version 3.02 software. Pathogen strains, which were previously inhibited on an agar diffusion assay were cultured to an OD_600nm_ of 0.2–0.25 or diluted with phosphate-buffered saline (PBS) to achieve this value [28]. In each microplate well, 100 µL of twofold BHI medium, 50 µL of BLIS, 48 µL of saline (NaCl 0.85%, *w*/*v*), and 2 µL of pathogenic strain suspension (1:100, *v*/*v*) were added. As a negative control (C−), no pathogenic cells were included, and, as a positive control (C+), no BLIS was used. The final volume of 200 µL was well-mixed and the microplate was incubated in the microplate reader. *C. perfringens*, a strict anaerobe bacterium, was grown under anaerobic conditions using an anaerobic jar containing a GasPak™ EZ Anaerobe Container System (BD™, Franklin Lakes, NJ, USA) and candle to accelerate oxygen depletion. In 1.5 mL microtubes, 200 µL of twofold thioglycolate medium (98 µL of medium, 100 µL of BLIS and 2 µL of *C. perfringens* suspension (1:200, *v*/*v*; grown overnight) was prepared. As C−, 2 µL of thioglycolate medium was used instead of pathogenic cells, and as a positive control, BLIS was replaced by thioglycolate medium. Twenty-five tubes were prepared (samples, C− and C+) to read samples and controls every hour for 24 h. Assays were performed in triplicate. 

### 2.4. Analysis of the BLIS Produced by L. lactis L2

Chromatographic analysis was performed on an Agilent 1260 Infinity II Preparative Liquid Chromatography System equipped with a G7115A diode detector (Agilent Technologies, Santa Clara, CA, USA) and a Persuit 5-C18 (4.6 × 250 mm, 5 μm). The mobile phase components were: (A) 0.1% Trifluoroacetic acid (TFA) in water and (B) 0.1% TFA in Liquid Chromatography/Mass Spectrometry (LC/MS) grade acetonitrile (Sigma Aldrich, Billerica, MA, USA). The column was equilibrated, and elution was performed using a linear gradient from 1% to 60% B over 60 min at a flow rate of 0.5 mL/min and temperature of 40 °C. The analytes were monitored at 220, 254, and 280 nm.

Subsequently, 10 μL of the fractions was analyzed separately, by ESI-Q-TOF mass spectrometry (Impact II Bruker Daltonics mass spectrometer, Billerica, MA, USA). The ESI-Q-TOF system was operated in the extracted ion mode, and chromatograms and full-scan MS spectra were acquired at a rate of 0.5 Hz. MS precursor and MS/MS product ions were acquired over the 100–4000 *m*/*z* range, and the collision-induced dissociation energy ranged from 7 to 70 eV [29].

### 2.5. In Silico Prediction of Bacteriocinogenic Production by L. lactis L2

To search in silico for bacteriocin genes phylogenetically related to *L. lactis* subsp. *lactis* L2, all non-anomalous complete genome assemblies of this subspecies were downloaded locally from the RefSeq database (37 sequences). Of these, all valid files were submitted to the BAGEL4 tool to predict bacteriocin genes and ribosomally synthesized and post-translationally modified peptides (RiPPs) and their associated neighboring proteins known to influence bacteriocin function. For the bacteriocin gene cluster, putative genomic loci containing transport/immunity genes, modification genes, leader cleavage peptides, and a structural peptide were considered for the analysis [30,31]. For these, genomic sequences were retrieved to calculate the MW of the core peptides in Daltons (Da) [32].

### 2.6. Statistical Analysis

All experiments were performed in triplicate and the numerical results generated were used to express the results. Means of replicates were calculated to express data, and standard deviations were calculated to express variance among experimental samples.

## 3. Results

### 3.1. Assessment of Antibacterial Activity against Pathogenic Strains

The BLIS produced by *L. lactis* L2 was unable to inhibit the growth of the Gram-negative bacteria *E. coli*, *S. enterica* serovar Choleraesuis and *S. enterica* serovar Typhimurium, but it was able to inhibit the growth of all Gram-positive bacteria tested. However, the growth inhibition was species-specific, with significant variation in the size of the diameter of the inhibition halos (Figure 1). According to the results, *E. faecalis* showed the greatest growth inhibition, presenting the best results in inhibition halo diameter (7.9 mm) and antimicrobial activity expressed by the BLIS concentration (19.69 AU/mL), followed by *C. perfringens* with growth inhibition of 6.9 mm and 13.98 AU/mL. The lowest growth inhibition was observed for *S. suis* (5.8 mm and 8.39 AU/mL), *S. hyicus* (5.2 mm and 5.8 AU/mL), and *S. dysgalactiae* (4.6 mm and 3.49 AU/mL) (Table 2).

In addition, the kinetic assay was performed to evaluate the growth of pathogenic strains after exposure to *L. lactis* L2 BLIS for 24 h (treatment), to determine whether the BLIS growth inhibition has a bacteriostatic or bactericidal effect. The absence of inhibitory growth of Gram-negative bacteria was confirmed, with similar growth kinetics represented by the cultivation of pathogenic bacteria in the presence of BLIS (treatment), in the absence of pathogenic bacteria (C+), and in the absence of pathogenic bacteria (C−) (Figure 2D,F). Gram-positive bacteria showed a shift in their growth kinetics compared to the C+. *E. faecalis*, *S. dysgalactiae*, and *S. hyicus* had a delay of 4–8 h in their growth, as shown in Figure 2A–C, respectively. These results showed that *L. lactis* L2 BLIS had a bacteriostatic effect on *E. faecalis*, *S. dysgalactiae*, and *S. hyicus*, presenting a bacteriostatic effect, since the lag phase of these pathogens was increased (11, 12, and 7 h, respectively) compared to the control (C+).

The growth of the *S. enterica* serovar Typhimurium, the *S. enterica* serovar Choleraesuis, and *E. coli* were not significantly inhibited (Figure 2D–F), and these results agree with the agar diffusion tests conducted previously (Figure 1, Table 2). The growth kinetics of *S. suis* and *C. perfringens* (Figure 2G and Figure 2H, respectively) demonstrated that *L. lactis* L2 BLIS was able to completely inhibit their growth throughout the growth kinetics, indicating a bactericidal effect of *L. lactis* L2 BLIS on these two pathogens.

### 3.2. Analysis of L. lactis L2 BLIS by High-Performance Liquid Chromatography (HPLC)

To identify the presence of antimicrobial peptides responsible for inhibiting the growth of pathogens, such as bacteriocins, the BLIS produced by *L. lactis* L2 was analyzed by HPLC based on its MW. Two types of analyses were performed: the first one was the analysis of the MRS medium to identify its specific spectrum; that is, to know which peaks correspond to this medium and exclude them from the BLIS analysis; and the second one was the BLIS analysis (Figure 3A). BLIS showed 12 peaks with different values of intensity and weight, ranging from 500 to 1160 Da. The peak with the highest intensity and concentration had a MW of 757.35 Da, while the peak with the lowest intensity but of higher MW was of 1157.788 Da (Figure 3B). When the peaks from MRS medium (Figure 3A) and BLIS (Figure 3B) were superimposed (Figure 3C), a significant difference was observed, with seven peaks from BLIS of higher intensity versus only one from MRS medium, with MWs ranging from 620 to 890 Da. Only one superimposition was observed, which was represented by the peaks of 889.43 (BLIS) and 893.55 (MRS medium), and these peaks were excluded from further analysis, resulting in a total of 6 peaks representing peptides contained in BLIS.

### 3.3. In Silico Prediction of Bacteriocin Genes

An in silico prediction was performed using data retrieved from 37 complete genome assemblies of *L. lactis* subsp. *lactis*, scored using the BAGEL4 software (v4). The in silico prediction of the BAGEL4 database indicated that bacteria from this subspecies are capable of producing eight bacteriocin genes within this dataset (Figure 4). We retrieved genomic sequences for each of these genes to relate them to scientific data. For the indicated bacteriocin genes shown in Figure 4, the MW varied from 5.68 to 29.95 kDa; much larger than what we have found in the BLIS of *L. lactis* L2, which ranged from 500–1160 Da.

## 4. Discussion

Bacteriocinogenic LAB are used as probiotics in several animal models, influencing the immune and inflammatory response, and, interestingly, are used for pathogen inhibition [15]. Probiotics are a promising alternative to reduce the use of antibiotics, which have been widely used for the last decades to control infections in animal production, seriously impacting antimicrobial resistance worldwide [33].

The BLIS produced by *L. lactis* L2 shows an inhibitory activity against most pathogens belonging to the Firmicutes phylum, which is highly prevalent in the human and animal gut microbiota [34,35]. Here, we describe its potential against cultured strains of *E. faecalis*, *S. suis*, *S. dysgalactiae*, *S. hyicus*, and *C. perfringens* isolated from diseased pigs. These opportunistic pathogens of the swine industry are found in oral, rectal, skin, and fecal sites of pigs and are responsible for infections such as diarrhea [36], meningitis, pneumonia [37], metritis, and cystitis [38].

A bacteriostatic effect induced by the BLIS of *L. lactis* on the growth of *E. faecalis*, *S. dysgalactiae*, and *S. hyicus* was observed after 15 h of cultivation in the presence of this BLIS. This is because it does not cause the death of bacteria, but alters their metabolic profile [39]. These results agree with those reported in the literature showing the inhibitory effects of different strain collections on these pathogens [40,41,42].

As a result of these experiments, the compounds present in the BLIS of *L. lactis* L2 have induced the complete inhibition of the growth of *S. suis* and *C. perfringens* in vitro, which is consistent with that reported in the literature of bacteriocinogenic mechanisms induced by LAB on 37 different pathogens [43,44,45], indicating the potential application for probiotic products.

It is important to highlight the ability of *C. perfringens* to form spores, which contributes to its lethality, resistance to stress factors, and impact in the food industry [46,47,48]. The BLIS produced by *L. lactis* L2 has shown great importance in the control of this pathogen in the food chain, avoiding its potential toxin in food products [46].

The BLIS tested did not induce growth inhibition when tested against Gram-negative strains in vitro. This resistance may be due to an outer membrane that blocks the action of antibiotics and other inhibitory molecules, making it urgent to develop new strategies to treat these infections and overcome antimicrobial resistance, as demonstrated by the resistance of *Salmonella* spp. and *E. coli* to the BLIS [49,50].

As it is known, bacteriocins may be applied to inhibit growth of related species, including particularly Gram-positive bacteria, as they have differences in cell wall composition and structure compared to Gram-negative bacteria [51]. Hence, these differences relate to their peptidoglycan layer in their cell wall, providing a target for bacteriocins, which can bind to it, attacking cell integrity and inducing cell death [52]. On the other hand, Gram-negative bacteria possess an additional outer membrane, providing another protection against bacteriocins. This acts as a permeability barrier, inhibiting adequate invasion of bacteriocins into the cell [53]. These biological features associate with our obtained results in vitro.

The BLIS produced by the *L. lactis* L2 strain was previously reported to carry the nisin gene in its DNA [21]. Nisin is a 3.4 kDa bacteriocin classified as a class I type A lantibiotic, which may be responsible for inducing the inhibition of various pathogens [50,54,55]. Therefore, we subsequently subjected this BLIS to HPLC analysis to determine whether it actually contained nisin or other peptides, as suggested by the bioinformatic analyses reported here. Nevertheless, none of the bacteriocins predicted by the in silico analysis were found to have MWs in the range of 500–1160 Da, as revealed by the HPLC analysis. In this sense, among the peptides with low MW, previously reported in the literature, are reported bacteriocins from LAB able to induce bacterial inhibition [25], such as mutacin from *S. mutans* of ~1000 Da [56], a non-pediocin peptide from *P. pentosaceus* strain IE-3 of 1701 Da [57], and a microcin produced by propionibacteria and enterobacteria weighing around 800–900 Da [58].

We propose that nisin is not the main antimicrobial compound produced by *L. lactis* responsible for bacterial inhibition. In this regard, the BLIS-induced inhibitory effect may also be a response to the presence of small peptides, as suggested by the results of the present work. Therefore, these non-nisin peptides, also present in the BLIS of *L. lactis* L2, could be potential candidates for the control of bacterial pathogen inhibition.

In this way, the application of antimicrobial peptides produced by LAB probiotic bacteria is a promising alternative for the control of bacterial pathogens in the pig industry, if it shows an inhibitory effect against relevant pig pathogens that cause direct economic and veterinary impacts [59]. These results reinforce the possibility of replacing the use of antibiotics with probiotics, which could potentially contribute as growth promoters, stimulating the growth of a healthy microbiota, preventing the colonization of enteric pathogens, producing antimicrobial compounds, and contributing to digestion and other health aspects [60,61,62]. After more than 50 years of widespread use of the former, growing concerns about their safety and contribution to antimicrobial resistance have prompted the search for healthier alternatives [63,64].

## 5. Conclusions

In this study, the effect of the BLIS produced by *L. lactis* subsp. *lactis* strain L2 was evaluated against bacterial swine pathogens, reporting the inhibitory potential of this LAB to be used against *C. perfringens*, *E. faecalis*, *S. dysgalactiae*, *S. suis*, and *S. hyicus*, which represent serious economic and health impacts to the pig industry. Despite these results, the BLIS tested was not effective against *S. enterica* serovar Typhimurium, S. *enterica* serovar Choleraesuis and *E. coli*. The analyses performed indicated the presence of molecules ranging from 500 to 1160 Da in the BLIS. This MW variation distinguishes the possible active molecules from the bacteriocins predicted in the *L. lactis* subsp. *lactis* genomes studied, which vary from 5.68 to 29.95 kDa. These results encourage the scientific community to search for new products and to develop alternatives for the prevention and control of pathogens in livestock.

## Figures and Tables

**Figure 1 animals-13-02442-f001:**
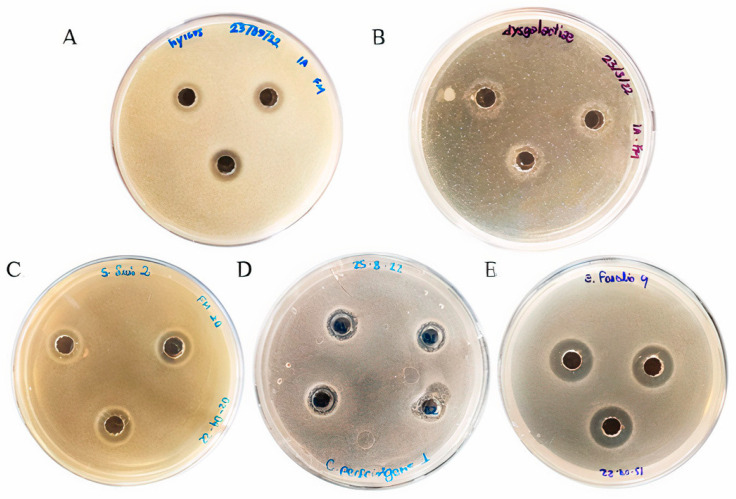
Pathogen growth inhibition after 24 h of incubation with BLIS produced by *L. lactis* L2. (**A**) *S. hyicus*; (**B**) *S. dysgalactiae*; (**C**) *S. suis*; (**D**) *C. perfringens*; (**E**) *E. faecalis*.

**Figure 2 animals-13-02442-f002:**
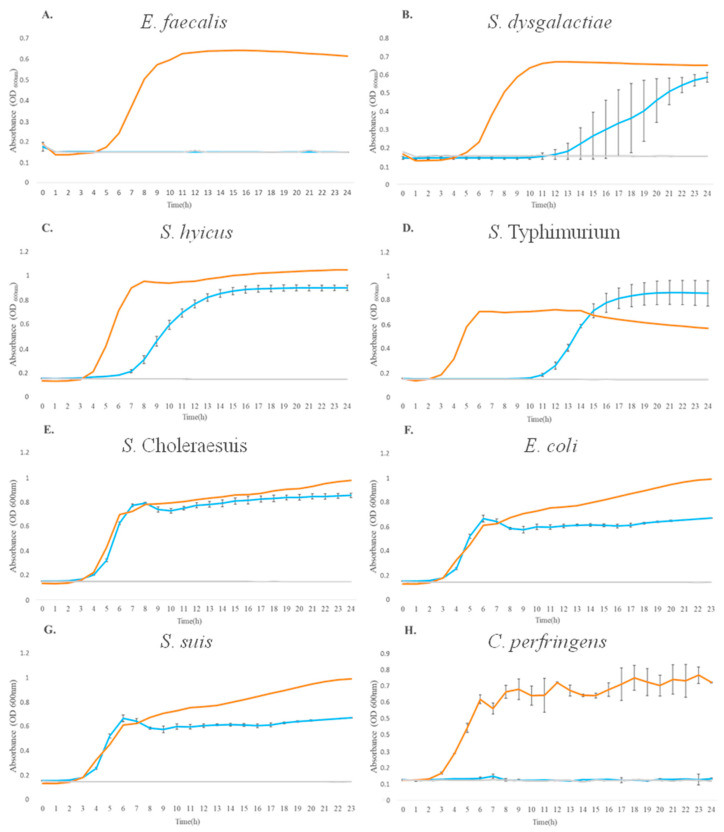
Growth kinetics of the pathogenic strains after 24 h of *L. lactis* L2 BLIS exposure. Blue line represents the growth of pathogenic strain in the presence of BLIS (treated); orange line represents the growth of pathogenic strain in the absence of BLIS (positive Control, C+); and gray line represents the absence of pathogenic strain (negative control, C−). Pathogens are, respectively, represented by: (**A**) *E. faecalis*; (**B**) *S. dysgalactiae*; (**C**) *S. hyicus*; (**D**) *S. enterica* serovar Typhimurium; (**E**) *S. enterica* serovar Choleraesuis; (**F**) *E. coli*; (**G**) *S. suis*; (**H**) *C. perfringens*.

**Figure 3 animals-13-02442-f003:**
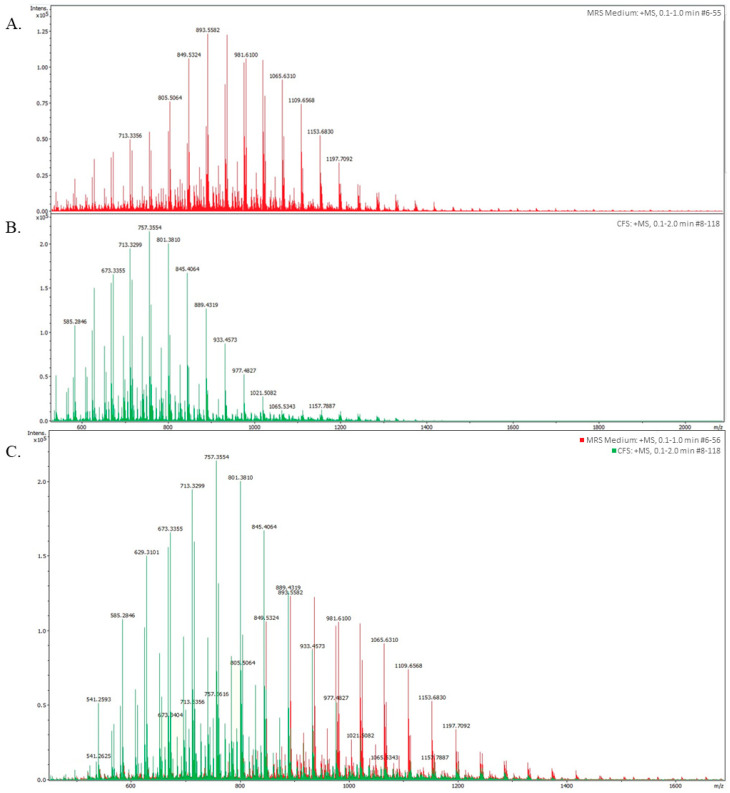
Molecular chromatogram of the BLIS produced by *L. lactis* L2 showing different peaks of intensity and MW. (**A**) components of the MRS medium; (**B**) components of the BLIS; (**C**) merge.

**Figure 4 animals-13-02442-f004:**
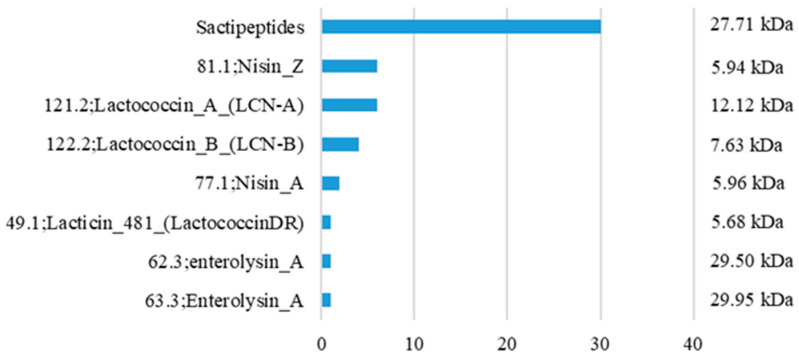
Prediction of bacteriocin genes within 37 *L. lactis* subsp. *lactis* genome assemblies and MW of their respective bacteriocins. Blue bars represent the number of hits for each bacteriocin identifier.

**Table 1 animals-13-02442-t001:** Pathogenic bacterial strains isolated from Brazilian pigs.

Pathogenic Bacteria	Isolation Site	Clinical Case	Age	State	Year
*Streptococcus suis*	Lung	Pneumonia	Nursery pig	São Paulo	2016
*Streptococcus dysgalactiae*	Brain	Meningitis	Nursery pig	Mato Grosso	2019
*Staphylococcus hyicus*	Skin	Pyoderma	Finishing pig	Santa Catarina	2013
*Enterococcus faecalis*	Spleen	Septicemia	Sow	São Paulo	2017
*Salmonella enterica* serovar Choleraesuis	Lung	Septicemia	Finishing pig	Paraná	2015
*Salmonella enterica* serovar Typhimurium	Feces	Enteritis	Sow	Santa Catarina	2012
*Escherichia coli*	Feces	Enteritis	Unweaned piglet	Rio Grande do Sul	2012
*Clostridium perfringens*	Feces	Enteritis	Unweaned piglet	São Paulo	2022

**Table 2 animals-13-02442-t002:** Inhibition effect of *L. lactis* L2 BLIS against Gram-negative and Gram-positive pathogenic strains. Results were expressed in diameter of inhibition halo (mm) and BLIS concentration by arbitrary unit per milliliter (AU/mL).

Pathogenic Strains	Area of Inhibition (cm^2^)	Diameter of Inhibition Halo (mm)	Antimicrobial Activity
*E. coli*	-	-	-
*S. enterica* serovar Typhimurium	-	-	-
*S. enterica* serovar Choleraesuis	-	-	-
*S. hyicus*	0.46	5.20	5.80
*S. dysgalactiae*	0.28	4.60	3.49
*S. suis*	0.67	5.80	8.39
*E. faecalis*	1.58	7.90	19.69
*C. perfringens*	1.11	6.90	13.88

## Data Availability

The data presented in this study are available on request from the corresponding author.

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
