# Peer review of "Antimicrobial Activity of Peptides Produced by Lactococcus lactis subsp. lactis on Swine Pathogens"

_animals, 2023, doi:10.3390/ani13152442_

Round 1
Reviewer 1 Report
In this manuscript from Sanca et al., the authors have investigated the effect of bacteriocin-like inhibitory substances (BLIS) of Lactococcus lactis subsp. lactis strain 37 L2 against Escherichia coli, Salmonella enterica, Streptococcus suis, Streptococcus dysgalactiae, Staphylococcus hyicus and Enterococcus faecalis, all pathogenic strains isolated from pig samples. The results obtained showed that BLIS had an inhibitory effect on Gram-positive bacteria, while no significant effect was reported against Gram-negative bacteria. HPLC analysis has shown that antimicrobial peptides with a molecular size of 500-1160 Daltons were isolated from BLIS.
Overall, the manuscript is fairly well-written, although I believe that the authors should address some issues listed below prior to resubmit their manuscript:
Introduction
L59-L60: -‘However, their use as growth promoters has been questioned due to the development and spread of antimicrobial resistant bacteria’.
I would say that the use/abuse of antibiotics has been questioned with regard to the control of diseases, other than growth promoters. The authors should change this sentence accordingly.
L70: ‘with good efficacy against Gram-positive bacteria’.
Is it just against Gram-positive? The authors should include also efficacy against Gram-negative. Please add more references.
L71-L72: Add references at the end of the sentence.
L81: Cholerasuis in Italic.
Material and Methods
2.2. Bacterial growth conditions
The names of bacteria must be in Italics and should be amended accordingly throughout the entire manuscript.
L119: in replicates for all pathogen strains. How many replicates? Please specify.
Results
Figure 2: I would add the name of the specific pathogen together or instead of the letters, to have an immediate understanding of the growth kinetics.
Discussion
According to their results, the Gram-positive bacteria were inhibited by the Lactobacillus while the Gram-negative were not. The authors should address better this in the Discussion section.
Author Response
ANSWERS TO THE COMMENTS OF REVIEWER #1
Answers to the comments of reviewer 1 are highlighted in blue throughout the manuscript
General comment
In this manuscript from Sanca et al., the authors have investigated the effect of bacteriocin-like inhibitory substances (BLIS) of Lactococcus lactis subsp. lactis strain L2 against Escherichia coli, Salmonella enterica, Streptococcus suis, Streptococcus dysgalactiae, Staphylococcus hyicus and Enterococcus faecalis, all pathogenic strains isolated from pig samples. The results obtained showed that BLIS had an inhibitory effect on Gram-positive bacteria, while no significant effect was reported against Gram-negative bacteria. HPLC analysis has shown that antimicrobial peptides with a molecular size of 500-1160 Daltons were isolated from BLIS.
Overall, the manuscript is fairly well-written, although I believe that the authors should address some issues listed below prior to resubmit their manuscript:
Answer: The authors appreciate the reviewer's comment. They also carefully reviewed the manuscript addressing the suggestions pointed by the reviewer.
Introduction
Comment 1. L59-L60: -‘However, their use as growth promoters has been questioned due to the development and spread of antimicrobial resistant bacteria’.
I would say that the use/abuse of antibiotics has been questioned with regard to the control of diseases, other than growth promoters. The authors should change this sentence accordingly.
Answer: The authors appreciate the reviewer's comment however, the authors would like to clarify that the statement mentioned above is in agreement with current data found in the literature. Accordingly, no changes were made to this sentence. Below are some current references on this subject.
- Vondruskova H. et al. (2010). Alternatives to antibiotic growth promoters in prevention of diarrhea in weaned piglets: a review. https://doi.org/10.17221/2998-VETMED
- Tracker P.A. (2013). Alternatives to antibiotics as growth promoters for use in swine production: a review. https://doi.org/10.1186/2049-1891-4-35
- Dowarah R. et al. (2017). The use of Lactobacillus as an alternative of antibiotic growth promoters in pig: A review. https://doi.org/10.1016/j.aninu.2016.11.002
- Cardinal K.M. et al. (2021). Estimation of productive losses caused by withdrawal of antibiotic growth promoter from pig diets – Meta-analysis. https://doi.org/10.1590/1678-992X-2020-0266
- Wen R. et al. (2022). Withdrawal of antibiotic growth promoters in China and its impact on the foodborne pathogen Campylobacter coli of swine. https://doi.org/10.3389/fmicb.2022.1004725
- Hughes P. & Heritage J. Antibiotic Growth-Promoters in Food Animals (https://www.adiveter.com/ftp_public/articulo1138.pdf)
Comment 2. L70: ‘with good efficacy against Gram-positive bacteria’.
Answer: The authors would like to thank the reviewer’s observation. This sentence was removed from the text.
Comment 3. Is it just against Gram-positive? The authors should include also efficacy against Gram-negative. Please add more references.
Answer: The authors would like to thank the reviewer’s comment however, information regarding the effectiveness of the BLIS produced by L. lactis L2 strain against Gram-negative pathogens has been described in the text (Lines: 182-184; 198-202; 283-287, 289-296 and Table 2).
Comment 4. L71-L72: Add references at the end of the sentence.
Answer: The authors would like to thank the reviewer’s observation. Proper references have been added in the paragraph (Lines 71-72).
Comment 5. L81: Cholerasuis in Italic.
Answer: The authors would like to thank the reviewer’s observation but serovar classifies an antigen variant and not the species, which means that the bacteria strain is distinguished from other strains by its antigenicity. This term refers to the bacteria infrasubspecific designation (below subspecies) and it is not written in italics. However, a careful review has been made of the entire text.
Material and Methods
Comment 6. 2.2. Bacterial growth conditions
The names of bacteria must be in Italics and should be amended accordingly throughout the entire manuscript.
Answer: The authors would like to thank the reviewer’s observation. The requested suggestion has been revised and corrected.
Comment 7. L119: in replicates for all pathogen strains. How many replicates? Please specify.
Answer: The requested specification has been added in the text.
Results
Comment 8. Figure 2: I would add the name of the specific pathogen together or instead of the letters, to have an immediate understanding of the growth kinetics.
Answer: The authors would like to thank the reviewer’s comment. The name of each pathogen has been added in their respective results in the figure.
Discussion
Comment 9. According to their results, the Gram-positive bacteria were inhibited by the Lactobacillus while the Gram-negative were not. The authors should address better this in the Discussion section.
Answer: The authors would like to thank the reviewer’s suggestion. This approach was improved in the Discussion section (Lines 289-296).
Reviewer 2 Report
The authors took up a very current topic, needed in pig farming. Unfortunately, the introduction of antibiotics into animals' bodies is currently a big problem in the meat industry. Later, this translates into antibiotic resistance of pathogenic bacteria and more and more cases of diseases that are difficult to treat. Bacteriocins have great potential and I am very happy that such research has been undertaken.
I have few comments on the text in only one chapter - "Materials and Methods".
Here are the notes:
lines 102 and 104 - how was the microaerophilic environment created? Refills or special dedicated sachets?
line 116 - The Authors state that "...OD600nm 0.2-0.3 was reached." Can this example be converted to colony forming units/mL?
line 117 - I don't understand the phrase "...at concentration of 1% concentration". What did the Authors mean here?
line 181 - statistical analysis, or rather the lack of it. In text is "...to express variance" but the analysis of variance wasn't there. It is a pity, because the results presented in Table 2 could be subjected to such an analysis.
Author Response
ANSWERS TO THE COMMENTS OF REVIEWER #2
Answers to the comments of reviewer 2 are highlighted in pink throughout the manuscript
The authors took up a very current topic, needed in pig farming. Unfortunately, the introduction of antibiotics into animals' bodies is currently a big problem in the meat industry. Later, this translates into antibiotic resistance of pathogenic bacteria and more and more cases of diseases that are difficult to treat. Bacteriocins have great potential and I am very happy that such research has been undertaken.
Answer: The authors sincerely appreciate the reviewer’s comment. They are gratified to find support and encouragement from their scientific peers.
I have few comments on the text in only one chapter - "Materials and Methods".
Here are the notes:
Comment 1. lines 102 and 104 - how was the microaerophilic environment created? Refills or special dedicated sachets?
Answer: The authors would like to thank the reviewer’s comment. In a microaerophilic environment, Lactococcus lactis L2 was grown in anaerobic jar using candle to eliminate internal oxygen. For anaerobic growth (i.e. C. perfringens), the airtight environment was created using GasPak™ EZ Anaerobe Container System (BD™, USA).
Comment 2. line 116 - The Authors state that "...OD600nm 0.2-0.3 was reached." Can this example be converted to colony forming units/mL?
Answer: The authors would like to thank the reviewer’s suggestion. The manuscript has been revised and the cell concentration was added to the text.
Comment 3. line 117 - I don't understand the phrase "...at concentration of 1% concentration". What did the Authors mean here?
Answer: The authors would like to apologize for this mistake. The statement was corrected to clarify that agar medium was supplement with agar (1%, w/v) to allow the solidification of the medium.
Comment 4. line 181 - statistical analysis, or rather the lack of it. In text is "...to express variance" but the analysis of variance wasn't there. It is a pity, because the results presented in Table 2 could be subjected to such an analysis.
Answer: The authors would like to appreciate the reviewer's comment. Our research focused on providing initial insights into bacterial inhibition in vitro against different swine pathogens through an exploratory investigation. Instead of developing an extensive statistical analysis, we employed a simpler approach using descriptive statistics, including replicates and mean values. This was made due to our research objectives and time constraints. Our results can futurely pave the way for more complex research in this field. Results presented in this article provide a baseline for understanding and guide future research. We value this feedback and will consider it for future studies, ensuring the inclusion of comprehensive analyses.
Reviewer 3 Report
The manuscript entitled “Antimicrobial activity of peptides produced by Lactococcus lactis subsp. lactis on swine pathogens” is well written and well structured.
Two considerations that the Authors should clarify:
1) The disk diffusion method for evaluating the inhibiting activity, although it takes into account the formula with which arbitrary units are established, is perhaps not the most suitable. It would have been better to use the MIC by determining it with the micromethod using microwell plates. What is the reason why the Authors preferred this method?
2) Of the strain of Escherichia coli used it would be useful to know at least the serogroup it belongs to.
In the conclusions chapter many of the terms referring to bacterial genus and species are not written in italics as they should be written. This carelessness also emerges in other parts of the manuscript. I advise the Authors to double check everything for this aspect.
When discussing the results, the Authors should compare the different activities found between Gram negatives and Gram positives, hypothesizing the reason for the differences also in the light of the scientific literature available on the subject.
Author Response
ANSWERS TO THE REVIEWER # 3
Answers to the comments of reviewer 3 are highlighted in green throughout the manuscript
General comment
The manuscript entitled “Antimicrobial activity of peptides produced by Lactococcus lactis subsp. lactis on swine pathogens” is well written and well structured.
Answer: The authors appreciate the reviewer's comment. They also carefully reviewed the manuscript addressing the suggestions pointed by the reviewer.
Two considerations that the Authors should clarify:
Comment 1. The disk diffusion method for evaluating the inhibiting activity, although it takes into account the formula with which arbitrary units are established, is perhaps not the most suitable. It would have been better to use the MIC by determining it with the micromethod using microwell plates. What is the reason why the Authors preferred this method?
Answer: This method was chosen because it is the most widely methodology used for first identification of inhibitory activity. The authors understand that MIC would present additional data considering the lowest inhibitory concentration of BLIS however, the authors opted for carrying out the methodology of inhibition kinetics, since it would determine whether the inhibitory effect of BLIS would be of a bacteriostatic or bactericidal nature.
Comment 2. Of the strain of Escherichia coli used it would be useful to know at least the serogroup it belongs to.
Answer: The authors would like to apologize for this failure. The characteristics of this E. coli strain have been added to the text (Line 109). However, the authors would like to clarify that it is not usual to perform the serotyping of E. coli strains related to cases of enteritis in swine, as the serogroup is not directly related to the virulence of the strain. For this reason, the data is not available. The E. coli strain studied belongs to the ETEC pathotype, which causes diarrhea in pigs. This strain was characterized for the presence of virulence genes related to this pathotype and, the result was positive for the presence of genes coding for the toxins STb, Sta, LT and fimbriae F4 and 987P.
Comment 3. In the conclusions chapter many of the terms referring to bacterial genus and species are not written in italics as they should be written. This carelessness also emerges in other parts of the manuscript. I advise the Authors to double check everything for this aspect.
Answer: The authors would like to apologize for this mistake. The entire text has been carefully revised and corrected however, for the genus Salmonella, the identification Choleraesuis and Typhimurium refer to the serotype and not the species, so they must remain with the first letter capitalized, without being in italics.
Comment 4. When discussing the results, the Authors should compare the different activities found between Gram negatives and Gram positives, hypothesizing the reason for the differences also in the light of the scientific literature available on the subject.
Answer: The authors would like to thank the reviewer’s suggestion. The discussion was improved considering this suggestion (Lines 289-296).
Reviewer 4 Report
The results may serve as for the industrial implementation, so potentially scientifically important findings. Also Authors do contribute in this field before, publishing on "Bacteriocinogenic probiotic bacteria isolated from an aquatic environment inhibit the growth of food and fish pathogens"
1. Please provide description of the Fig. 2 F, G, H in the legend section. Not clear what does it refer to, although the description can be found in the manuscript text.
2. Point 3.2. please comment on the chromatogram peaks possibly referring to other peptides than BLIS origin. It's not necessarily only BLIS present in the supernatant, many other proteins or peptides can be secreted from the cell into the bacteriological medium. Please indicate which can solely be identified as BLIS. Comment also on 12 mentioned peaks versus 8 genes coding for the BLIS components found in this species. Degradation? different peptides? contamination? peptide-peptide chemical interaction?
1. Please correct any misspellings, including lack of italics in the names of bacterial species (e.g. line 43, etc.).
Author Response
ANSWERS TO THE REVIEWER # 4
Answers to the comments of reviewer 4 are highlighted in yellow throughout the manuscript
General comment
The results may serve as for the industrial implementation, so potentially scientifically important findings. Also Authors do contribute in this field before, publishing on "Bacteriocinogenic probiotic bacteria isolated from an aquatic environment inhibit the growth of food and fish pathogens"
Answer: The authors sincerely appreciate the reviewer’s comment. They are gratified to find support and encouragement from their scientific peers.
Comment 1. Please provide description of the Fig. 2 F, G, H in the legend section. Not clear what does it refer to, although the description can be found in the manuscript text.
Answer: The authors would like to thank the reviewer’s observation. The missing information has been added in the caption of the Figure 2.
Comment 2. Point 3.2. please comment on the chromatogram peaks possibly referring to other peptides than BLIS origin. It's not necessarily only BLIS present in the supernatant, many other proteins or peptides can be secreted from the cell into the bacteriological medium. Please indicate which can solely be identified as BLIS. Comment also on 12 mentioned peaks versus 8 genes coding for the BLIS components found in this species. Degradation? different peptides? contamination? peptide-peptide chemical interaction?
Answer: The authors would like to thank the reviewer’s suggestion. But the total of obtained peaks was six, due to a previous selection where the peaks with the greatest spectrum were selected, and one of them was discarded because it showed the same molecular weight as a spectrum of the MRS culture medium. In order to identify these peptides, it was used the BAGEL4 database, being found eight different core peptides associated to bacteriocins in genomes of L. lactis subsp. lactis, but none of them with molecular weight similar to those found through HPLC in our experiments. Therefore, in the discussion, we suggest that those are new peptides of low molecular weight inhibiting that bacteria tested in this research.
Comments on the Quality of English Language
Comment 1. Please correct any misspellings, including lack of italics in the names of bacterial species (e.g. line 43, etc.).
Answer: The authors would like to thank the reviewer’s observation. The entire text has been carefully revised.
Round 2
Reviewer 3 Report
I believe that the manuscript properly corrected and supplemented by the authors as suggested can be published.